# Molecular, Morphological and Chemical Diversity of Two New Species of Antarctic Diatoms, *Craspedostauros ineffabilis* sp. nov. and *Craspedostauros zucchellii* sp. nov.

Riccardo Trentin [1,2], Emanuela Moschin [1], André Duarte Lopes [2,3], Stefano Schiaparelli [4,5], Luísa Custódio [2,*] and Isabella Moro [1,*]

1 Department of Biology, University of Padova, 35131 Padova, Italy
2 Centre of Marine Sciences, Faculty of Sciences and Technology, University of Algarve, 8005-139 Faro, Portugal
3 Department of Chemistry and Pharmacy, Fundação para a Ciência e a Tecnologia, University of the Algarve, 8005-039 Faro, Portugal
4 Department of Earth, Environment and Life Sciences, University of Genoa, 16132 Genova, Italy
5 Italian National Antarctic Museum (MNA, Section of Genoa), University of Genoa, 16132 Genova, Italy
* Correspondence: lcustodio@ualg.pt (L.C.); isabella.moro@unipd.it (I.M.); Tel.: +351-93-8272926 (L.C.); +39-049-8276255 (I.M.)

**Abstract:** The current study focuses on the biological diversity of two strains of Antarctic diatoms (strains IMA082A and IMA088A) collected and isolated from the Ross Sea (Antarctica) during the XXXIV Italian Antarctic Expedition. Both species presented the typical morphological characters of the genus *Craspedostauros*: cribrate areolae, two "fore-and-aft" chloroplasts and a narrow "stauros". This classification is congruent with the molecular phylogeny based on the concatenated 18S rDNA-*rbc*L-*psb*C alignment, which showed that these algae formed a monophyletic lineage including six taxonomically accepted species of *Craspedostauros*. Since the study of the evolution of this genus and of others raphe-bearing diatoms with a "stauros" is particularly challenging and their phylogeny is still debated, we tested alternative tree topologies to evaluate the relationships among these taxa. The metabolic fingerprinting approach was implemented for the assessment of the chemical diversity of IMA082A and IMA088A. In conclusion, combining (1) traditional morphological features used in diatoms identification, (2) phylogenetic analyses of the small subunit rDNA (18S rDNA), *rbc*L and *psb*C genes, and (3) metabolic fingerprint, we described the strains IMA082A and IMA088A as *Craspedostauros ineffabilis* sp. nov. and *Craspedostauros zucchellii* sp. nov. as new species, respectively.

**Keywords:** Antarctica; biodiversity; Ross Sea; diatoms; *Craspedostauros*; *Craspedostauros ineffabilis* sp. nov.; *Craspedostauros zucchellii* sp. nov.; morphology; molecular phylogeny; chemical diversity

## 1. Introduction

Traditionally, the taxonomic identification of diatoms mostly relied on the morphological features of the frustule [1] followed by other characters, such as ultrastructure, sexual reproduction and the development of auxospores [2]. Recently, the advent of molecular tools and "omics" sciences has provided more reliable methods for the study of the molecular and chemical diversity of microalgae and seaweeds [1–3]. The combination of morphological, molecular and chemical data, referred to as the polyphasic or integrative approach [4,5], does not only provide an insight into the evolutionary relations among diatom species, but could also deliver useful information on the functioning of diatoms [1]. This study aims to evaluate the different shades of photosynthetic biodiversity (morphological, ecological, molecular and chemical diversity) of two species of diatoms of the genus *Craspedostauros* E.J. Cox collected and isolated from the Ross Sea (Antarctica) during the XXXIV Italian Antarctic Expedition. *Craspedostauros* is a relatively small genus of raphid pennate diatoms, characterized by the "stauros", which is an internal transverse rib of silica

at the center of the valve [6,7]. This genus was distinguished by Cox, 1999 from *Stauroneis* Ehrenberg, according to the following morphological features: (1) cribrate areolae, (2) plate-like plastids with central pyrenoids, H-shaped in valve view, located "fore-and-aft" of the center of the cell, and (3) a central fascia (a transverse hyaline area of the valve) wider than the associated stauros [6,7]. The addition of novel molecular data to the traditional systematic tools has shed light on the evolution and the diversity of the genus *Craspedostauros*, with the description of novel species and the taxonomic revision of several genera [6,8]. However, while morphological data demonstrated the non-monophyletic nature of the stauros across raphid diatoms [7,9], in 2017, Ashworth et al. [6] highlighted several problems with the phylogenetic position of the genera *Craspedostauros* and *Staurotropis* T.B.B. Paddock by testing if the topology implied by a morphology-based tree was significantly different from that recovered from analysis of a DNA based tree. We followed this approach performing a phylogenetic analysis of the concatenated 18S rDNA, *rbc*L and *psb*C gene alignment based on the molecular dataset used by Ashworth et al. 2017 [6] expanded by including the sequences of *Craspedostauros danayanus* Majewska & Ashworth and *Craspedostauros macewanii* Majewska & Ashworth [8] and those generated in this study from the Antarctic diatoms strain IMA082A and IMA088A. Moreover, we assessed the robustness of trees constraining the monophyly of: (1) Mastogloiales: *Craspedostauros*, *Achnanthes* Bory, and *Mastogloia* Thwaites ex W.Smith (CAM); (2) quadrate rotae taxa: *Craspedostauros*, *Staurotropis*, *Achnanthes* and *Mastogloia* (CSAM) and the genera *Craspedostauros* and *Achnanthes* (CA), which were reported as sister taxa by Ashworth et al. 2017 [6].

In this study, the untargeted UPLC-HR-MS/MS metabolomic approach was applied for the detection of key biomarkers that allowed to distinguish two closely related species of Antarctic diatoms. Metabolome is currently used in a wide array of research fields and particularly, metabolic fingerprinting has been shown to be a promising method in the classification and taxonomy of filamentous fungi, yeast and microalgae [1,10–12]. Secondary metabolites profiling has a high differentiation at order, genus and species levels in fungi [11] seaweeds [3] and microalgae [13] and was proposed as a novel method to differentiate northern and southern strains of the cryptic diatom *Chaetoceros socialis* reinforcing morphological and molecular data [1]. In this sense, we applied metabolic fingerprinting in the taxonomic study of cryptic *Craspedostauros* species isolated from the Antarctica during the XXXIV Italian Antarctic Expedition (2018/2019). Strains IMA082A and IMA088A collected from two different sites in Antarctica characterized by the same environmental conditions, cultivated at the same conditions and harvested during the exponential growth phase. Algal extracts were subsequently analyzed by liquid chromatography-mass spectrometry (LC-MS) and the differences among the two strains were visualized by principal component analysis based on their metabolic profiles and variations in metabolite markers were visualized through hierarchical clustering heat maps.

Finally, we describe two novel species of Antarctic diatoms, namely *Craspedostauros ineffabilis* sp. nov. strain IMA082A and *Craspedostauros zucchellii* sp. nov. strain IMA088A through this integrative approach. Specifically, to support the establishment of this lineages, we provide (1) morphological data, acquired with light and scanning electron microscopy (SEM), (2) a phylogeny of the concatenated 18S rDNA, *rbc*L and *psb*C gene alignment and (3) metabolic fingerprinting.

## 2. Materials and Methods

### 2.1. Isolation and Cultures

During the XXXIV Italian Antarctic Expedition (2018/2019), from samples collected in Terra Nova Bay, Ross Sea, Antarctica by Isabella Moro, two diatom strains were isolated: IMA082A from a seawater sample collected from Inexpressible Island with coordinates 74°54′ S 163°39′ E, and IMA088A obtained from a sampling of the desalination plant filters of the Italian research Mario Zucchelli Station, with coordinates 74°41′39.06″ S 164°07′18.18″ E (Figure 1). The diatoms strains IMA082A and IMA088A were grown both

in F/2 [14] growth medium with a salinity of 35‰ in a growth chamber at 5 °C and a light intensity of 10 μmol photons m$^{-2}$ × s$^{-1}$.

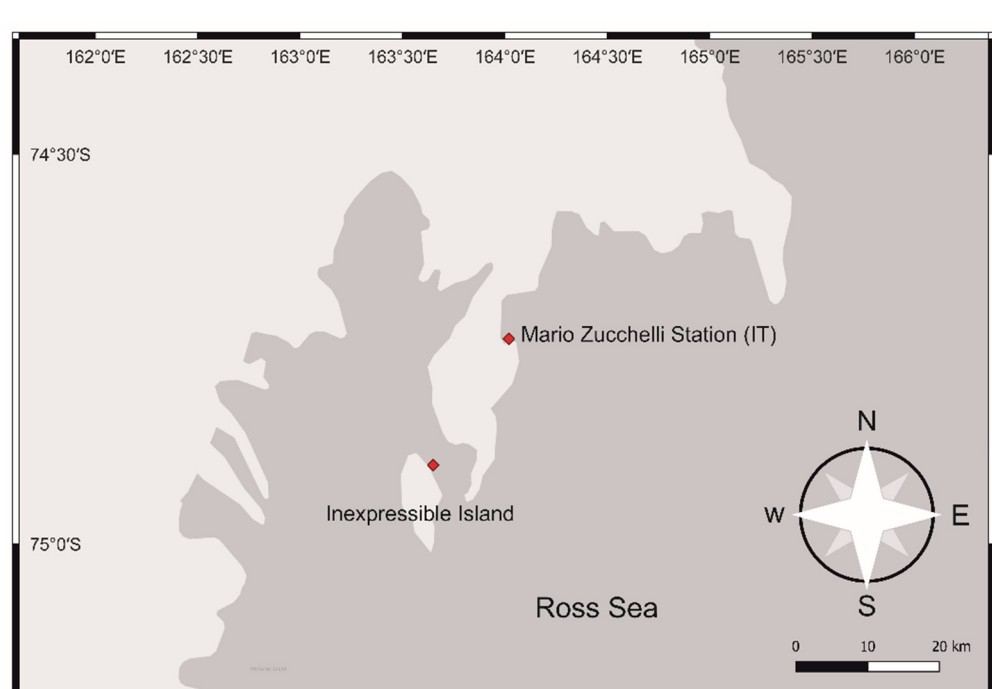

**Figure 1.** Map of the sampling sites, indicated by red diamonds.

### 2.2. Molecular Analysis

2.2.1. DNA Extraction and Amplification of Selected Molecular Markers

Culture aliquots of IMA082A and IMA088A isolates were harvested and centrifuged for 5 min at 13,000× $g$ in a Sigma 1–15 microcentrifuge (Sigma Laborzentrifugen GmbH, Germany). Pelleted cells were ground in a mortar with a pestle using quartz sand. The homogenate was recovered and the DNA was extracted using the DNeasy Powersoil Pro Kit® (Qiagen, Hilden, Germany), following manufacturer's indications. Genomic DNA was quantified with a DU 530 Beckman Coulter UV/vis spectrophotometer (Beckman Instruments Inc., Fullerton, CA, USA). Three molecular markers (18S rDNA, *rbcL* and *psbC* partial gene sequences) were amplified for phylogenetic analysis. The 18S rDNA region was amplified using the general eukaryotic primers Euk528F, EukA, EukB, Euk1209F and U1391R [15–19] and the amplification profiles used by Hugerth et al. 2014 [20]. Partial fragments of *rbcL* and *psbC* were amplified using the primer pairs *rbcL*1255+/*rbcL*66− and *psbC*+/*psbC*− [21] with the amplification profile described by Alverson et al. 2007 [21]. PCR products were verified by gel electrophoresis and purified with HT ExoSAP-IT High-Throughput PCR Product Cleanup reagent (ThermoFisher Scientific, Waltham, MA, USA) before sequencing. Sequencing was performed by the BMR Genomics Sequencing Service (University of Padua) using the same amplification primers used for 18S rDNA and *rbcL*, while the primer pair *psbC*221+/*psbC*857 [21] was used for the sequencing of *psbC*. The final consensus sequences, created using SeqMan II from Lasergene package (DNASTar, Madison, Wisconsin, USA), were compared with those available in online databases by using BLAST tool [22].

The obtained sequences for IMA082A (1432 for the 18S rDNA gene, 1055 bp for the *rbcL* gene, 714 bp for the *psbC* gene) and IMA088A (1630 bp for the 18S rDNA gene, 896 bp for the *rbcL* gene, 866 bp for the *psbC* gene,) were deposited in GenBank, with the following accession numbers: OP354221, OP354493 and OP354495 (18S rDNA, *rbcL* and *psbC* for IMA082A) and OP354222, OP354494 and OP354496 (18S rDNA, *rbcL* and *psbC* for IMA088A).

2.2.2. Phylogenetic Analyses

Separate alignments were created for 18S rDNA, *rbcL* and *psbC* using sequences obtained in this study and other sequences of diatoms included in the dataset used by Ashworth et al. 2017 were implemented with new sequences of *Craspedostauros* retrieved from the GenBank® database [23] (Table S1). 18S rDNA sequences were aligned using the program SSUalign [24], *rbcL* and *psbC* datasets were aligned with CLUSTALW [25] implemented in MEGA-X 10.2.4 [26]. The resulting alignments were concatenated in MEGA-X 10.2.4 and a concatenated phylogenetic tree was inferred by maximum likelihood criteria using IQTree 1.6.12 [27]. ModelFinder [28] was used to find the best evolutionary models for each of the three subsets (18S rDNA, *rbc*L and *psb*C). The best models were selected by the Akaikie Information Criterion. For the 18S rDNA partition, a TIM3+F+I+G model with empirical base frequencies (F), a proportion of invariable sites (I) and a gamma shape parameter (G) was adopted, while for both the *rbc*L and *psb*C partitions a GTR+F+I+G model with empirical base frequencies (F), proportion of invariable sites (I) and a gamma shape parameter (G) was chosen. The robustness of the topologies was assessed by approximate Likelihood Ratio Tests (aLRT) based on Shimodaira–Hasegawa (SH)-like procedures [29] in IQTree, by Bayesian inference (BI) in MrBayes v3.2.7 [30] and bootstrap (BT) re-samplings (1000 replicates) in IQTree. The BI analyses consisted of two separate concurrent Markov chain Monte Carlo (MCMC) runs, each composed of four chains (three heated and one cold), for $5 \times 10^6$ generations, with trees sampling every 1000 generations. The posterior distribution at the end of each run was considered adequate if the average standard deviation of the split frequencies was $\leq 0.01$. The first 12,500 trees were discarded as burn-in, as determined with Tracer version 1.7 [31], and the consensus topology and posterior probabilities (PP) were derived from the remaining trees. Different topologies of the concatenated tree were tested in IQTree using the RELL method [32], Kishino–Hasegawa test [33], Shimodaira–Hasegawa test [34], expected likelihood weights [35] and approximately unbiased (AU) test [36] with 10,000 resamplings.

*2.3. Morphological Analysis*

Samples of IMA082A and IMA088A were examined using a Leitz Dialux 22 light microscope (Leitz, Westlar, Germany) equipped with an Optika C-P8 digital camera (Optika, Bergamo, Italy). Scanning electron microscope (SEM) observations were carried out using a scanning electron microscope FEI Quanta 200 variable pressure-environmental/ESEM (FEI, Eindhoven, The Netherlands) at a working distance of 7.5–9.8 mm and at a 20 kV voltage. IMA082A and IMA088A cells were prepared for SEM microscopy by fixation in glutaraldehyde 2.5% in 0.1 M cacodylate buffer (pH 6.9) and post-fixation in 1% osmium tetroxide (OsO4), in the same buffer, for 2 h. Samples were dehydrated in a graded concentration increasing ethanol series with centrifugation ($1500 \times g$ for 3 min) following every different concentration step. Subsequently, ethanol was removed by critical point drying and samples were gold-coated by sputtering for 4 min with a Sputter Coater (Edwards, Crawley, UK). Morpho-ecological data for strains IMA082A, IMA088A and other *Craspedostauros* species were summarized in a matrix and visualized using a principal component map calculated with the 'PCAmixdata' package implemented in R-Statistics® 3.5.3 version (The R Foundation, Vienna, Austria).

*2.4. Metabolomics*

2.4.1. Sample Preparation

Briefly, microalgal biomass was harvested at exponential growth and lyophilized. Subsequently, 50 mg of lyophilized biomass of IMA082A and IMA088A were extracted using acetone 80% (*v/v*) and cell disruption by bead-beating in an MM400 mixer mill (Retsch, Haan, Germany) at 30 Hz for 5 min. The extracts were centrifuged at $12,000 \times g$ for 10 min at 4 °C and the supernatants were collected. After the recovery of the supernatants, the remaining biomass was re-extracted until both the pellet and the supernatant became colorless. The extracts were incubated overnight at 4 °C and subsequently

filtered (Whatman no. 4) and dried under reduced vacuum pressure at 40 °C. Dried extracts were weighed, dissolved in pure methanol at the concentration of 10 mg/mL, filtered (0.2 nm) and diluted at the concentration of 2 mg/mL for the determination of their chemical profiles.

### 2.4.2. UPLC-HR-MS/MS Profiling of Extracts

The chemical profiles of IMA082A and IMA088A extracts were analyzed by liquid chromatography-high resolution mass spectrometry (LC-HR-MS) following the procedure described by Silva et al. 2022 [13]. A Thermo Scientific™ UltiMate™ 3000 UHPLC, coupled with an Orbitrap Elite (Thermo Fisher Scientific, Waltham, MA, USA) mass spectrometer with a Heated Electro-Spray Ionization source (HESI-II; Thermo Scientific) was used to analyze the extracts. 5 µL of each extract (1:10 diluted in pure LC-MS grade methanol) were injected and separated using a Thermo Scientific Accucore RP-18 column (2.1 × 100 mm, 2.6 µm) running for 40 min. The mobile phase consisted of ultra-pure LC-MS grade water with 0.1% formic acid and LC-MS grade acetonitrile, containing 0.1% formic acid. Positive and negative polarity data were acquired in separate runs. Extracts were analyzed in data-dependent mode with the selection of the three most intense ions under dynamic exclusion and collision-induced dissociation (CID) activation. MS/MS fragmentation was achieved using 35 keV rising collision energy in an isolation window of 2. The minimum signal required for ddMS2 triggering was 1000. Xcalibur v4.1 Qual Browser (Thermo Scientific, Waltham, MA, USA) was used for LC-MS data acquisition and subsequent analyses. Three independent replicates per each extract were used.

### 2.4.3. Metabolomic Data Processing and Statistical Analysis

Raw data were processed using Compound Discoverer™ 3.2.0.421 (Thermo Fisher Scientific, Waltham, MA, USA). Untargeted metabolomics workflow (Untargeted Metabolomics with Statistics Detect Unknowns with ID Using Online Databases and mzLogic) was used to perform retention time alignment and unknown compound identification using publicly available databases. The "Detect Unknown Compounds" node parameters included default values with exception for mass tolerance (set to 10 [ppm]) and min. peak intensity (set to 1000). The "Search ChemSpider" node was used to search mass spectral databases for matching compounds within a specified mass tolerance range or with a certain elemental composition using Natural Products Atlas [37], Lipid Maps [38], KEGG [39], Drugbank [40], Carotenoids Database [41], Human Metabolome Database [42], Phenol Explorer [43] and BioCyc [44] online databases. Furthermore, the following mass lists included in the Compound Discoverer software were searched: Arita Lab 6549 flavonoid structure database, EFS HRAM compound database, Endogenous Metabolites database 4400 compounds, Lipid Maps Structure database, Natural Product Atlas. A blank was used for background subtraction and noise removal during the pre-processing step. The "cleaned-up" feature lists (see Tables S2 and S3) were used to perform multivariate analysis on the metabolomics profiles of IMA082A and IMA088A extracts.

Initially, principal component analysis (PCA) was carried out in Compound Discoverer to investigate clustering patterns in the dataset. Subsequently, ANOVA test on normalized signals of each identified metabolite was carried out in Compound Discoverer to evaluate the statistical differences ($p$-value < 0.005) among the metabolomes of IMA082A and IMA088A. A hierarchical clustering of the statistically significant metabolites was implemented in R-Statistics® 3.5.3 version.

## 3. Results

### 3.1. Phylogenetic Analysis and Hypothesis Testing

In the best unconstrained tree (Figure 2) recovered from the concatenated (18S rDNA-*rbc*L-*psb*C) alignment, the Antarctic strains IMA082A and IMA088A belonged to a well-supported clade (98/1/100) that also included sequences of 6 *Craspedostauros* species described as *C. alatus* Majewska & M.P. Ashworth, *C. amphoroides* (Grunow ex A.W.F.

Schmidt) E.J.Cox, *C. alyoubii* J. Sabir & M.P. Ashworth, *C. paradoxus* M.P. Ashworth & Lobban, *C. danayanus*, *C. macewanii* [6–8,45]. Strain IMA082A recovered as the sister taxon to *C. amphoroides* with strong statistical support (100/1/100), while strain IMA088A was sister taxon to the clade formed by IMA082A and *C. amphoroides* with high statistical support (100/1/100). The results of the phylogenetic analyses were assessed to evaluate the robustness of different topologies. Trees constraining the monophyly of all Mastogloiales: *Craspedostauros*, *Achnanthes*, and *Mastogloia* (CAM) and quadrate rotae taxa: *Craspedostauros*, *Staurotropis*, *Achnanthes* and *Mastogloia* (CSAM), were significantly rejected, while the tree constraining to monophyly *Craspedostauros* and *Achnanthes* (CA) was not significantly excluded (Table 1). Our phylogram showed unequivocally that the polar strains IMA082A and IMA088A belong to separate species, which are closely related.

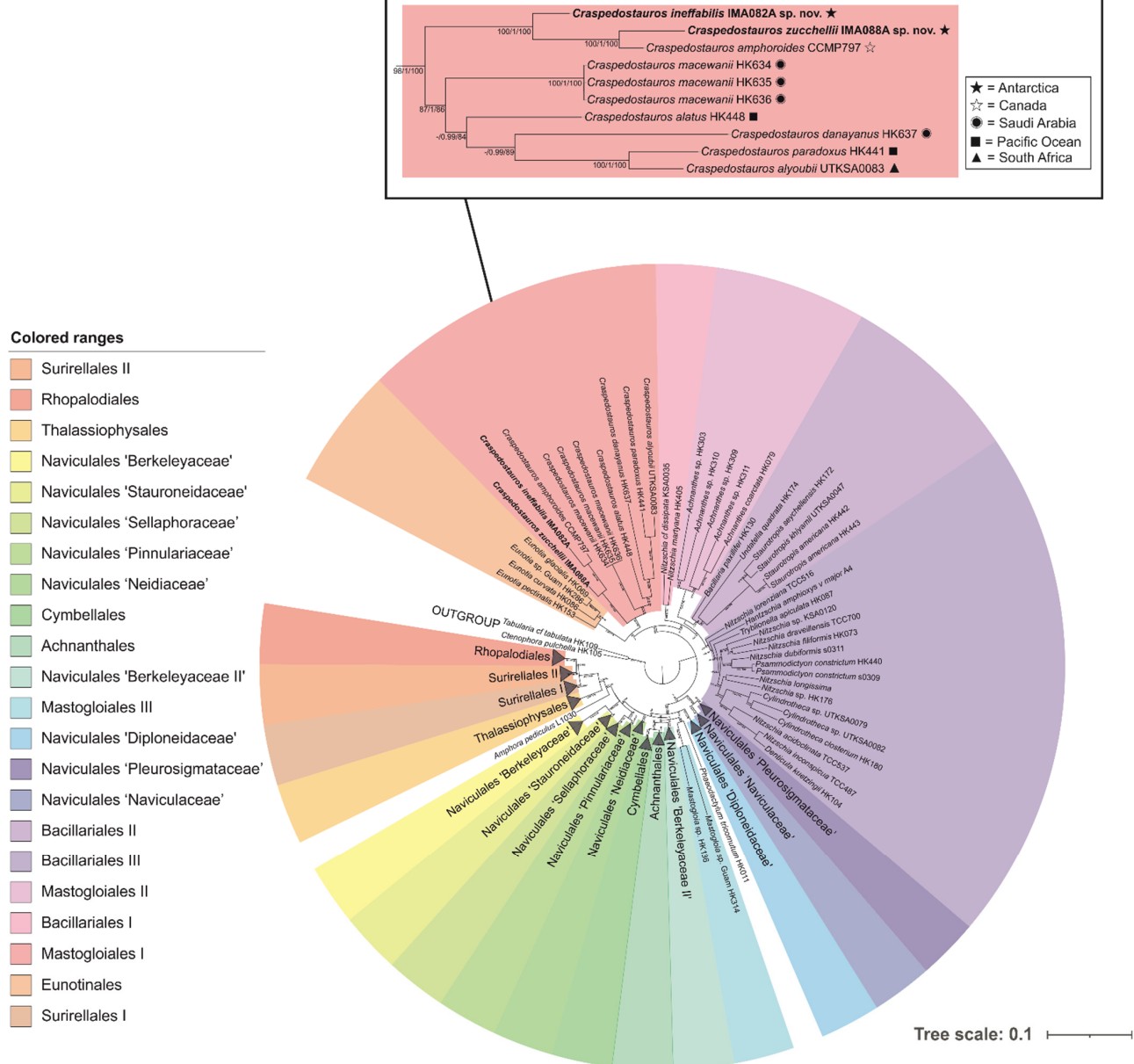

**Figure 2.** Maximum likelihood (ML) phylogeny of the concatenated 18S rDNA, *rbc*L and *psb*C regions alignment with members of the genus *Craspedostauros*. Approximate likelihood ratio tests based on Shimodaira–Hasegawa-like procedures (SH-aLRT) values (%), Bayesian posterior probabilities (PP) and ML bootstrap values (%) are shown above branches (SH-aLRT supports $\geq$ 80%, posterior probabilities $\geq$ 0.70 and bootstrap values $\geq$ 50%).

**Table 1.** Results of topology test. Scores of constrained trees: (BT) = Best unconstrained tree, (CA) = *Craspedostauros* and *Achnanthes*, (CAM) = *Craspedostauros*, *Achnanthes*, and *Mastogloia* and (CSAM) = *Craspedostauros*, *Staurotropis*, *Achnanthes* and *Mastogloia*. Plus signs denote the 95% confidence sets. Minus signs denote significant exclusion. All tests performed 10,000 resamplings using the RELL method.

| Tree | logL | deltaL [1] | bp-RELL [2] | *p*-KH [3] | *p*-SH [4] | *p*-WKH [5] | *p*-WSH [6] | c-ELW [7] | *p*-AU [8] |
|------|------|-----------|------------|-----------|-----------|------------|------------|-----------|-----------|
| BT | −51.868 | 0 | 0.646+ | 0.642+ | 1+ | 0.642+ | 0.862+ | 0.638+ | 0.627+ |
| CA | −51.870 | 2 | 0.354+ | 0.358+ | 0.712+ | 0.358+ | 0.646+ | 0.362+ | 0.373+ |
| CAM | −51.945 | 77 | 0− | 0− | 0− | 0− | 0− | $4.04 \times 10^{-12}$− | $1.94 \times 10^{-07}$− |
| CSAM | −51.977 | 108.99 | 0− | 0− | 0− | 0− | 0− | $1.05 \times 10^{-18}$− | $3.04 \times 10^{-47}$− |

[1] deltaL: logL difference from the maximal logl in the set. [2] bp-RELL: bootstrap proportion using RELL method (Kishino et al. 1990). [3] *p*-KH: *p*-value of one-sided Kishino–Hasegawa test (1989). [4] *p*-SH: *p*-value of Shimodaira–Hasegawa test (2000). [5] *p*-WKH: *p*-value of weighted KH test. [6] *p*-WSH: *p*-value of weighted SH test. [7] c-ELW: Expected likelihood weight (Strimmer & Rambaut 2002). [8] *p*-AU: *p*-value of approximately unbiased (AU) test [36].

### 3.2. Morphological Observations and Taxonomic Descriptions

Through light and scanning electron microscope observations, strains IMA082A and IMA088A appeared as pennate raphid diatoms, as well as our results based on DNA data.

The valve of IMA082A is linear elongate with rounded apices and a central stauros (Figure 3a). Two lobed, H-shaped plastids are present; one each between the central area and valve apex (Figure 3a). The auxospore is spherical (Figure 3b). Internally, the central area of valve shows a very slightly raised stauros, narrow and broadens at the valve center (Figure 3c) and central raphe endings terminate onto helictoglossae (Figure 3c). At the poles the external raphe fissure deflects strongly to one side. Valves show indistinct valve face-mantle junction (Figure 3d–f). In girdle view the cells have a slightly central constriction so the valve margin is not markedly biarcuate and curve slightly at the poles (Figure 3a,d–f). The girdle view shows many pored bands (Figure 3d–f). Areolae are similar in size throughout the valve, squarish to rounded, and broadening abruptly close to the raphe (Figure 3g,h). Striae are uniseriate, 25–36 in 10 μm, parallel through all the valve length (Figure 3c–h). Areolae are externally occluded by cribra perforated by 4–7 pores (Figure 3g,h).

The valve of IMA088A is linear to lanceolate in shape, with rounded apices and a central stauros (Figure 4a–c). The raphe slit is straight, curving strongly and unilaterally at the poles. Two lobed, H-shaped plastids are present; one each between the central area and valve apex (Figure 4a). The auxospore is spherical (Figure 4b). In girdle view the valve margin is not markedly biarcuate with a slight constriction in the center and curving slightly at the ends (Figure 4a,d). Striae are uniseriate, 22–31 in 10 μm, parallel through all the valve length (Figure 4c,d). Valves with distinct valve face-mantle junction (Figure 4d). Mantle has many rows of pored girdle bands (Figure 4d). Rounded areolae increase in size close to the raphe (Figure 4e,f). Internally, the stauros is slightly raised (Figure 4g) and central raphe endings terminate onto helictoglossae (Figure 4g). Areolae are occluded by cribra perforated by four to more pores (Figure 4f,h). Morpho-ecological data of IMA082A, IMA088A and other species of the genus *Craspedostauros* (Table 2) were visualized in a principal component map (Figure 5). The first component accounted for 10.17% of the total variance among species, while the second component accounted for 9.34% of the variance.

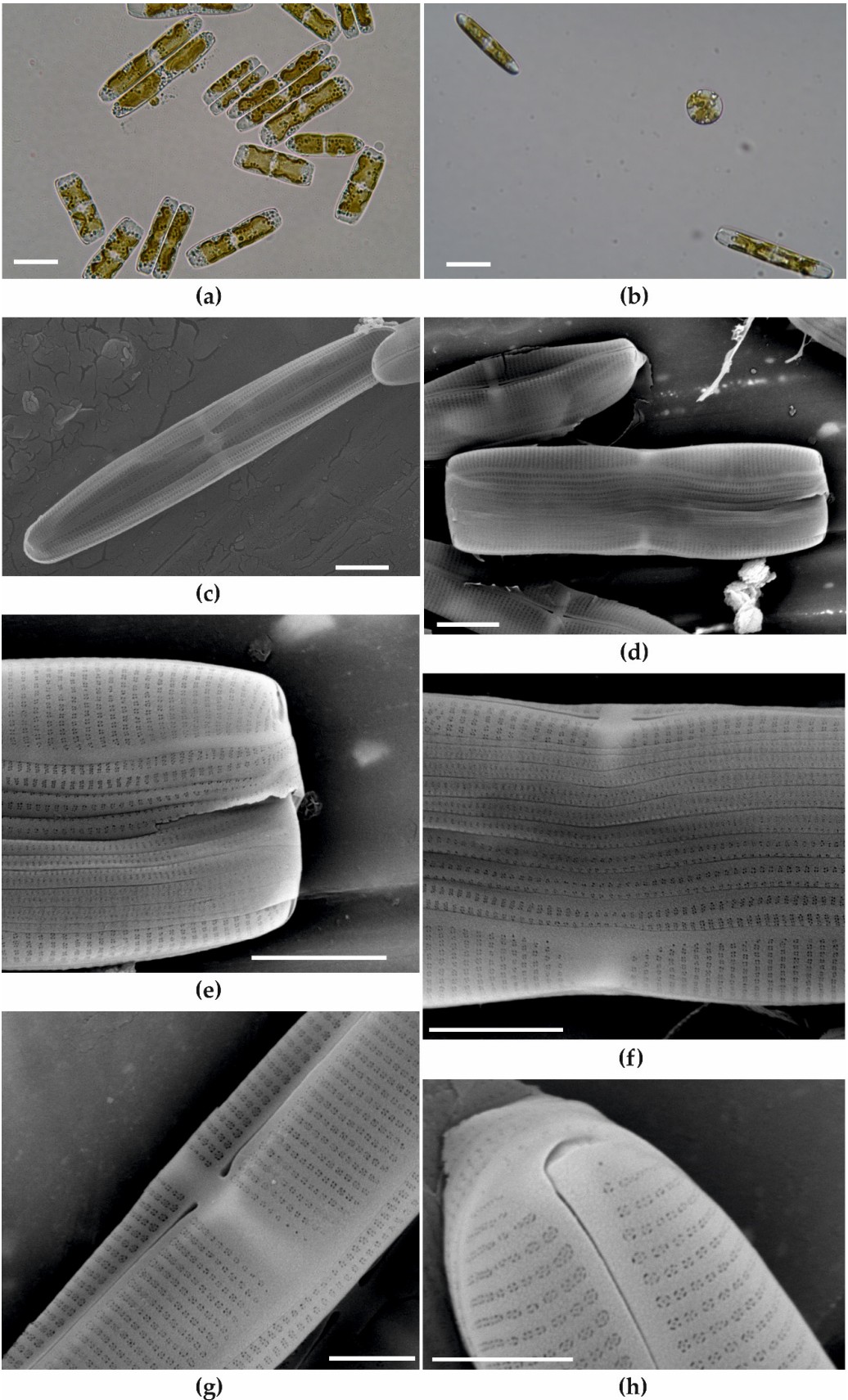

**Figure 3.** Light (**a**,**b**) and scanning electron images (**c**–**h**) of *Craspedostauros ineffabilis* IMA082A. Scale bars: (**a**,**b**) = 20 µm; (**c**,**d**) = 5 µm; (**e**,**f**) = 4 µm; (**g**,**h**) = 2 µm.

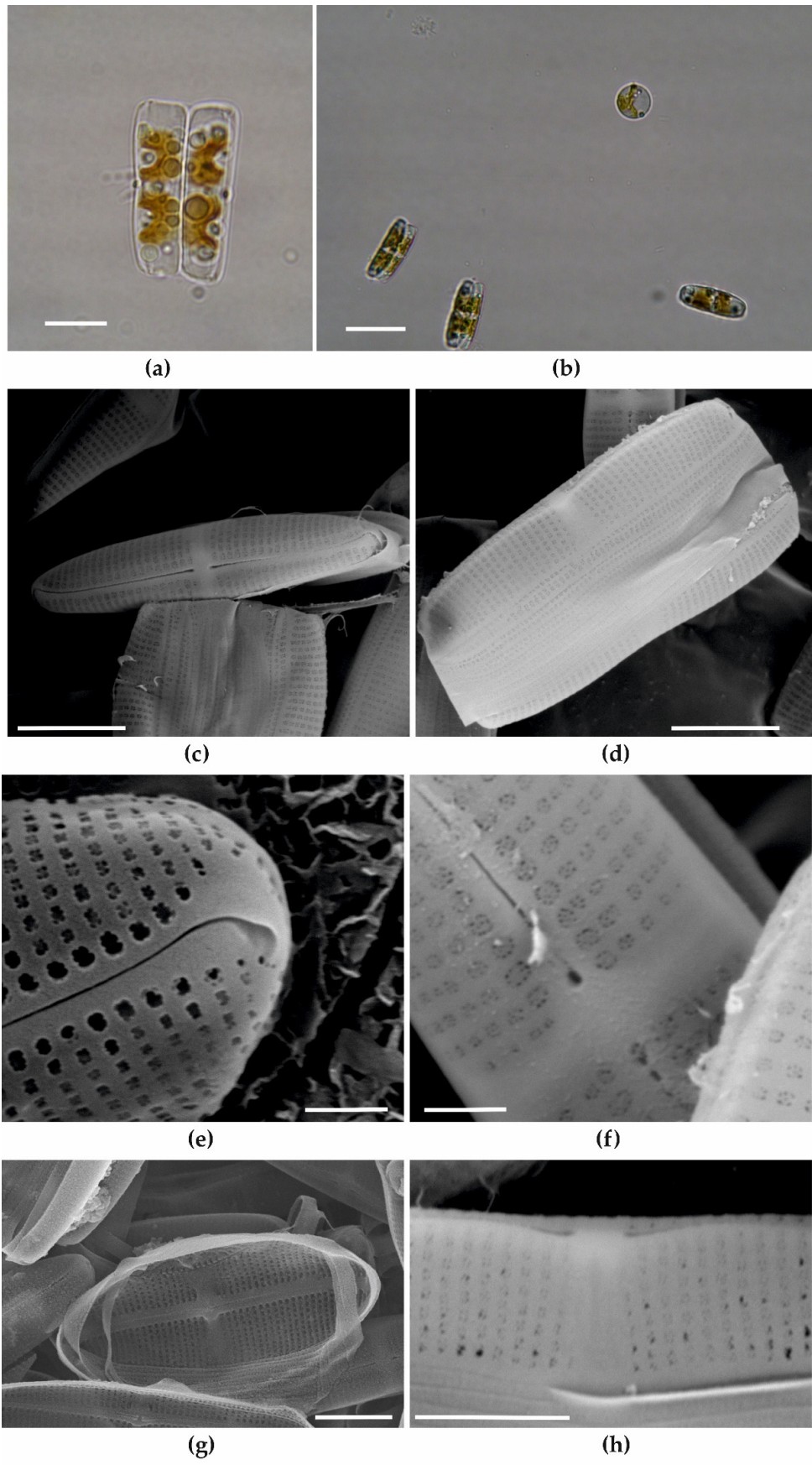

**Figure 4.** Light (**a**,**b**) and scanning electron images (**c**–**h**) of *Craspedostauros zucchellii* IMA088A. Scale bars: (**a**) = 10 μm; (**b**) = 20 μm; (**c**,**d**,**g**) = 5 μm; (**e**,**f**,**h**) = 1 μm.

**Table 2.** Morphological comparison of *Craspedostauros* species.

| | Valve Outline | Valve Length (μm) | Valve Width (μm) | Stria Density (in 10 μm) | Areola Size | Internal Central Area | Valve Face–Mantle Junction | Valve Margin at Centre | Average Number of Areolae Pores | Central Lip-Like Silica Flaps | Type Locality | Habitat | Reference |
|---|---|---|---|---|---|---|---|---|---|---|---|---|---|
| *C. neoconstrictus* | ±Linear, constricted | 40–110 | 5–7 | ~25 | Similar | Slight helictoglossae | Indistinct | Expanded | 6 (?) | Unknown | Sussex, England | Marine | [7] |
| *C. decipiens* | Lanceolate | 20–38 | 3–5 | 20–22 | Variable | Unknown | Distinct | Straight | 10–12 | Unknown | Bosporus | Marine | [7] |
| *C. capensis* | Lanceolate, constricted | 25–35 | 4.5–5.5 | 19 | Variable | Rectelevatum + Knob | Indistinct | Straight | 5–13 | Rudimentary | Cape Columbine, South Africa | Marine (intertidal) | [7] |
| *C. britannicus* | Linear to narrow lanceolate | 14–60 | 5–6 | ~24 | Similar | Helictoglossae | Indistinct | Slightly expanded | 5(+) | Rudimentary | Seascale, Cumbria, UK | Marine (driftwood) | [7] |
| *C. australis* | Linear | 35–78 | 4–6 | 35 | Similar | Rectelevatum + Knob | Indistinct | Straight | 4 | Rudimentary | Port Phillip Bay, Australia | Marine | [7] |
| *C. amphoroides* | Lanceolate to slightly constricted | 28–45 | 3.5–7 | 30–32 | Variable | Slight helictoglossae | Distinct | Straight | Unknown | Unknown | Frauenfeld's marin Aquarium | Marine | [7] |
| *C. alyoubii* | Linear, slightly-constricted | 83–105 | 6–10 | ~40 | Similar | Rectelevatum + Knob | Indistinct | Straight | 4–5 | Prominent | Duba, Saudi Arabia | Marine | [6] |
| *C. danayanus* | Linear, very slightly constricted | 28–61 | 2–2.5 | 49–51 | Similar | Rectelevatum | Indistinct | Straight | 6–8 | Absent | Mabibi Beach, Elephant Coast, South Africa | Marine (carapace of a sea turtle) | [8] |
| *C. legouvelloanus* [a] | Linear to linear-lanceolate, slightly constricted | 18–34 (23–39) | 3–5 (–6) | 46–49 (40–44) | Similar | Rectelevatum + knob with central cavity | Indistinct | Clearly expanded | 4 | Well developed | Kosi Bay, South Africa | Marine (carapace of a sea turtle) | [8] |
| *C. macewanii* | Linear to linear-lanceolate, slightly constricted | 26–51 | 4.5–5.5 | 28–31 | Similar | Rectelevatum + knob | Distinct | Straight | Highly variable | Rudimentary | uShaka Sea World, Durban, South Africa | Marine (carapace of a sea turtle) | [8] |
| *C. paradoxus* | Linear, slightly-constricted | 80–85 | 6.5–9 | 36–40 | Similar; can be longer near valve edge | Rectelevatum + Knob | Indistinct | Straight | 4–5 | Prominent | Gab Gab reef, Apra Harbor Guam, USA | Marine | [6] |
| *C. alatus* [a] | Linear to linear-lanceolate, slightly constricted | 20–37 (16–38) | 3–5 (5–7) | 26–28 (22–25) | Variable | Rectelevatum | Distinct | Very slightly expanded | Highly variable | Rudimentary | Riverhead, New York, USA | Marine (carapace of a sea turtle) | [45] |
| *C. laevissimus* | Linear to linear-lanceolate, without consction | 37–66 | 5.5–7.8 | 24–27.5 | Similar | Helictoglossae | Unknown | Unknown | 3–6 | Unknown | Lakes and pools in the Larsemann Hills, Rauer Islands and Bølingen Islands, Antarctica | Brackish water | [46] |
| *C. laevissimus* (*Tropidoneis laevissima*) | Linear to linear-lanceolate, without consction | 27–79 | 5–9 | 20–28 | Unknown | Unknown | Unknown | Unknown | Unknown | Unknown | Lakes and ponds in Kasumi Rock and on Shin-nan Rock, Antarctica | Brackish water | [47,48] |

**Table 2.** *Cont.*

| | Valve Outline | Valve Length (µm) | Valve Width (µm) | Stria Density (in 10 µm) | Areola Size | Internal Central Area | Valve Face–Mantle Junction | Valve Margin at Centre | Average Number of Areolae Pores | Central Lip-Like Silica Flaps | Type Locality | Habitat | Reference |
|---|---|---|---|---|---|---|---|---|---|---|---|---|---|
| *C. laevissimus* | Linear to Eliptical | 49–98 | 8–9.5 | Absent | Unknown | Unknown | Unknown | Unknown | Unknown | Unknown | Clear Lake, Green Lake, and lake on west side of McMurdo Sound, Antarctica | Fresh and Brackish water | [49] |
| *C. indubitabilis (Stauronella indubitabilis)* | Linear to eliptical | 25–60 | 6–7 | 25–27 | Unknown | Unknown | Unknown | Unknown | Unknown | Unknown | Arctic Ocean | Marine | [50] |
| *C. indubitabilis (Stauroneis constricta)* | Lanceolate, sometimes slightly constricted | 30 | 5 | 23–27 | Similar | Rectelevatum + Knob | Unknown | Unknown | 4–5 | Unknown | – | – | [51] |
| *C. indubitabilis* | Linear to eliptical, sometimes slightly constricted | 16–27 | 4.4–7.6 | 22–30 | Variable | Rectelevatum + Knob | Unknown | Unknown | 3–13 | Rudimentary | Bahía Salada, Caldera, Chile | Marine costal water | [52] |
| *C. infeffabilis* | Linear | 31–58 | 4–7.5 | 25–36 | Similar | Helictoglossae | Indistinct | Straight | 4–7 | Absent | Inexpressible Island, Terra Nova Bay, Antarctica | Marine costal water | This study |
| *C. zucchellii* | Linear to lanceolate | 14.5–24 | 3–6 | 22–31 | Variable | Helictoglossae | Distinct | Straight | 4(+) | Absent | Zucchelli Station, Terra Nova Bay, Antarctica | Marine costal water | This study |

[a] Values and descriptions given in brackets refer to the Adriatic populations.

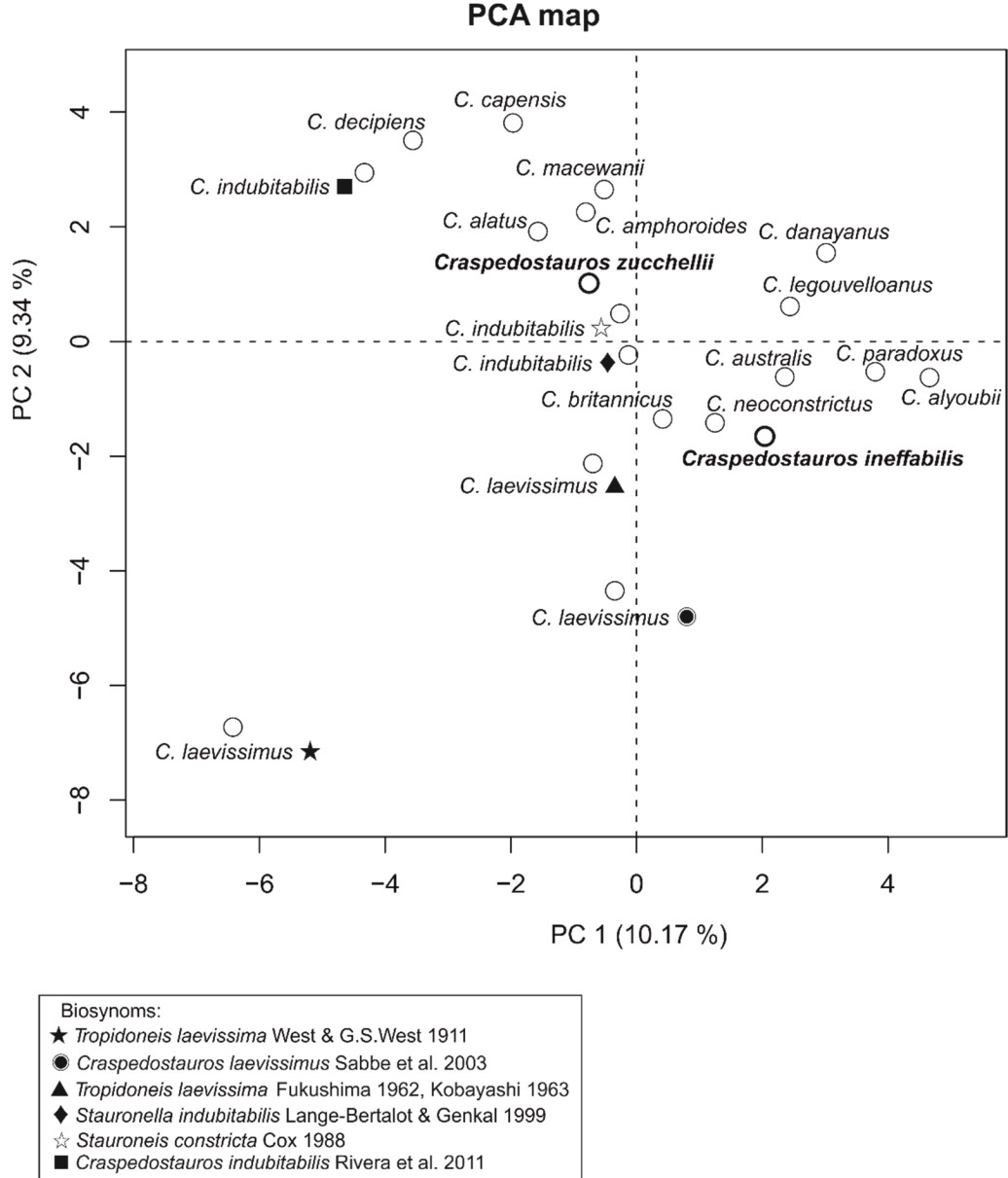

**Figure 5.** Principal component map showing morphological and ecological dissimilarities between *Craspedostauros* species [46–51,53].

Molecular phylogeny, together with morpho-ecological data supports the idea that the Antarctic strains IMA082A and IMA088A collected during the XXXIV Italian Antarctic Expedition belong to two new species of diatoms of the genus *Craspedostauros*. In addition to the molecular data, discussed in the section above, we provide morphological observations of both strains to formally describe these new taxa.

*Craspedostauros ineffabilis* Trentin, Moschin, Lopes, Custódio & Moro sp. nov.

DIAGNOSIS: Cells elongate without central constrictions and slightly tapering to the apices. Length 31–58 μm, width 4–7.5 μm; transapical striae 25–36 in 10 μm. Areolae are quadrate rota of similar size with 4–7 pores. Two H-shaped plastids, one on each side of the valve center. Silica flaps absent. Stauros present across the central area. Helictoglossae terminate the internal central raphe fissures. Rectelevatum internal central raphe ending.

HOLOTYPE: Strain IMA082A fixed culture and permanent slide deposited in the Italian National Antarctic Museum (MNA, Section of Genoa) with the voucher code MNA-15098 and MNA-15099, respectively.

TYPE LOCALITY: Inexpressible Island (Penguin Lagoon, Terra Nova Bay, Ross Sea, Antarctica), coordinates: 74°54′ S 163°39′ E.

ETYMOLOGY: The epithet 'ineffabilis' derives from the Latin, which refers to the name of the Island, where this new species was collected.

REFERENCE DNA SEQUENCES: GenBank accession numbers OP354221 (18S rDNA), OP354493 (*rbc*L) and OP354495 (*psb*C).

COMMENTS: *Craspedostauros ineffabilis* shows morphological resemblance to *C. laevissimus* (West & G.S. West) K. Sabbe but differs in salt tolerance. *C. ineffabilis* might be conspecific with *C. laevissimus* (West & G.S. West) from lakes and pools in the Larsemann Hills, Rauer Islands and Bølingen Islands. However, their different salt tolerance supports the idea that their belong to separate species, following the ecological definition of species. To overcome this issue, the sequencing of *C. laevissimus* is required.

*Craspedostauros zucchellii* Trentin, Moschin, Lopes, Custódio & Moro sp. nov.

DIAGNOSIS: Cells elongate without central constrictions and slightly tapering to the apices. Length 14.5–24 μm, width 3–6 μm; transapical striae 22–31 in 10 μm. Areolae are quadrate rota of variable size, larger near the raphe and with more pores more pores in the cribrum. Two H-shaped plastids, one on each side of the valve center. Silica flaps absent. Stauros present across the central area. Helictoglossae terminate the internal central raphe fissures. Rectelevatum internal central raphe ending.

HOLOTYPE: Strain IMA088A fixed culture and permanent slide deposited in the Italian National Antarctic Museum (MNA, Section of Genoa) with the voucher code MNA-15100 and MNA-15101, respectively.

TYPE LOCALITY: Italian Research Base Mario Zucchelli Station; coordinates 74°41′39.06″ S 164°07′18.18″ E.

ETYMOLOGY: The epithet name 'zucchellii' is dedicated to Mario Zucchelli, an Italian engineer and researcher, president of the ENEA Consortium for Antarctica, who devoted himself to development of the Italian Research in Antarctica.

REFERENCE DNA SEQUENCES: GenBank accession numbers OP354222 (18S rDNA), OP354494 (*rbc*L) and OP354496 (*psb*C).

COMMENTS: *Craspedostauros zucchellii* differs from *C. ineffabilis* for dimensions (length and width) and for areola size.

### 3.3. Liquid Chromatography-Mass Spectrometry (LC-MS)-Based Metabolomics Analysis

The untargeted UPLC-HR-MS/MS approach was used to explore the metabolite profiles of IMA082A and IMA088A extracts. Chromatograms from full scan measurements showed a high similarity between biological replicates, suggesting a high reproducibility and reduced biological variations between replicates (Figures S1 and S2). IMA082A and IMA088A extracts formed two distinct clusters, as shown in the principal components analysis (PCA) based on their metabolite profiles in both positive and negative ion modes (Figure 6a,b). The first component accounted for 68.7% and 69.3% of the variance among replicates, for positive and negative ion mode, respectively, while the second component accounted for 15.1% and 12.2% of the variance. This indicates a clear separation of metabolic signatures among the two strains of Antarctic diatoms. Among the 325 filtered features on the LC-HRMS metabolic fingerprints, 66 compounds in positive ion mode and 36 in negative ion mode presented a *p*-value lower than 0.005 among the two diatom species indicating significant differences of their metabolomic profiles (Figures 7 and 8). The main group of biomarkers that distinguished IMA082A from IMA088A were fatty acyls (18% in positive and 50% in negative ion mode). Other biomarker families identified by the UPLC-HR-MS/MS approach were amino acids and peptides, aromatic secondary metabolites, isoprenoids, alkaloids and polyketides.

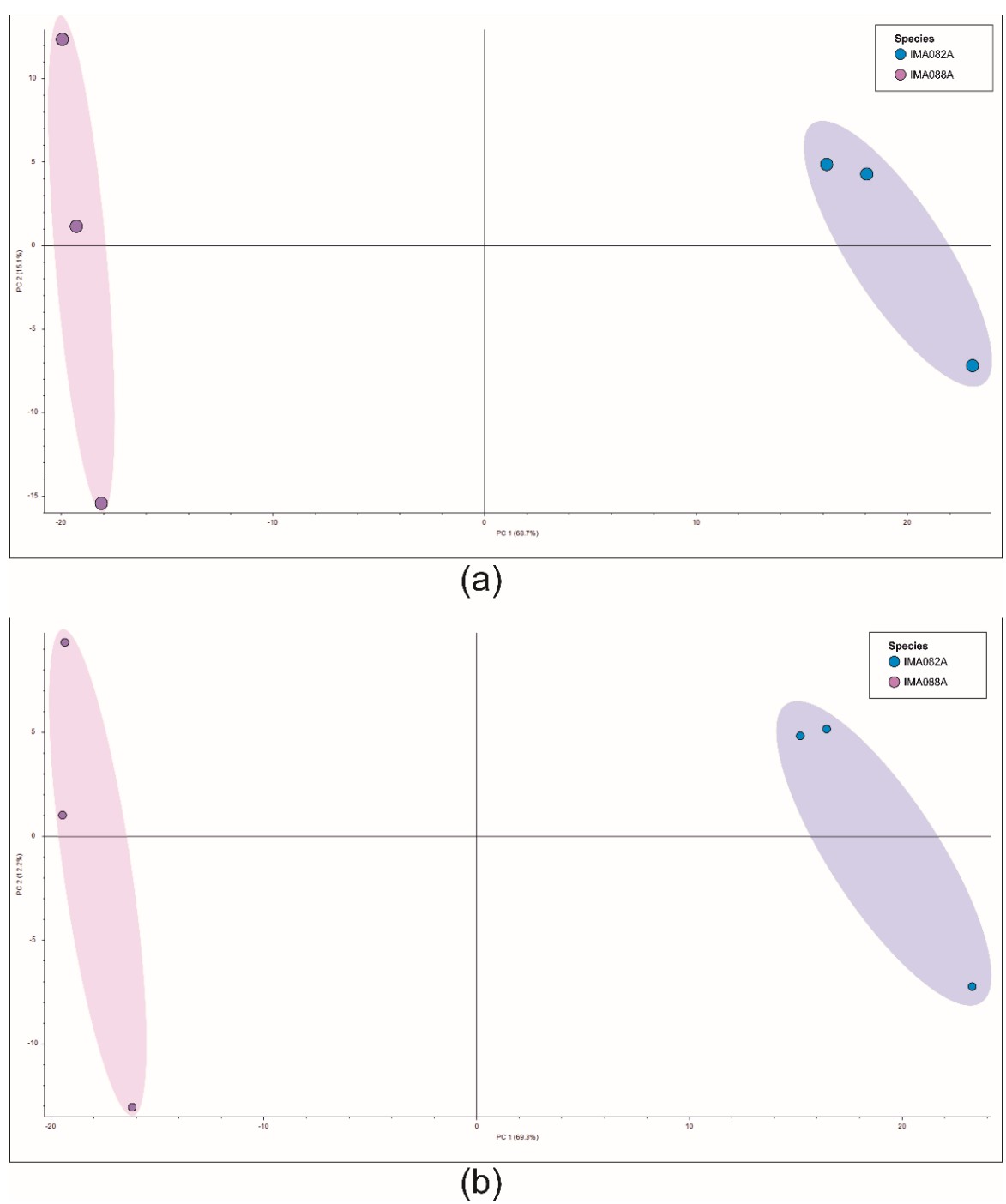

**Figure 6.** Principal components analysis (PCA) of metabolomic profiles of *C. ineffabilis* IMA082A and *C. zucchellii* IMA088A. The x- and y-axes represent principal components 1 and 2, respectively, while the percentages in brackets indicate how much of the overall variance in each dataset is explained by each principal component. Chemical profiles of three independent replicates were analyzed per *Craspedostauros* strain. (**a**) Positive ion mode; (**b**) Negative ion mode.

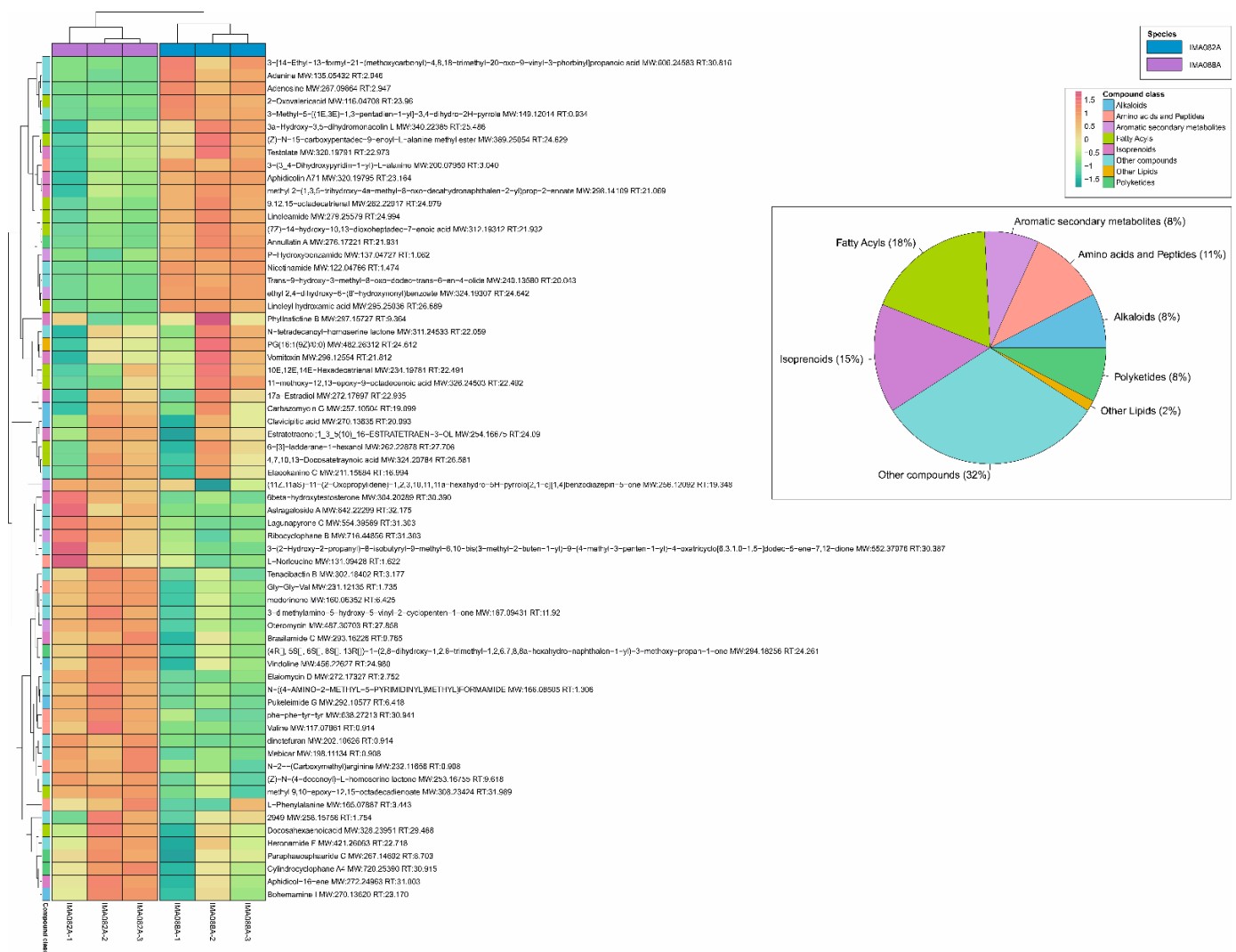

**Figure 7.** Feature analyses for candidate biomarker selection (*p*-values < 0.005). Hierarchical clustering heat map of metabolites between groups in positive ion mode. Colors from red to green indicate the normalized relative abundance values of metabolites from low to high according to the scale bar. Samples of *C. ineffabilis* IMA082A (blue) and *C. zucchellii* IMA088A (violet) are disposed in columns; replicates are indicated with suffix numbers 1, 2 and 3. Compound classes are indicated in different colors and percentages of each class are reported on the right side of the figure.

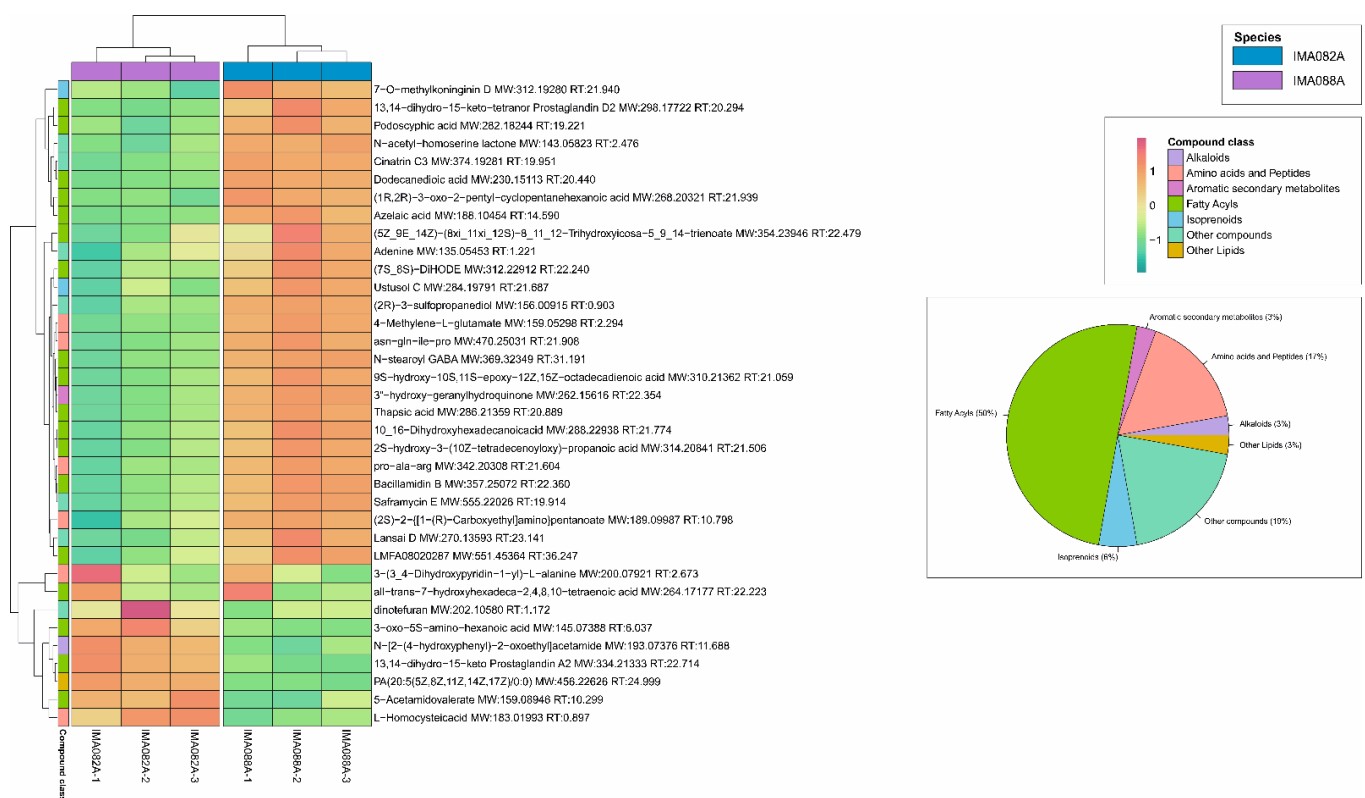

**Figure 8.** Feature analyses for candidate biomarker selection (*p*-values < 0.005). Hierarchical clustering heat map of metabolites between groups in negative ion mode. Colors from red to green indicate the normalized relative abundance values of metabolites from low to high according to the scale bar. Samples of *C. ineffabilis* IMA082A (blue) and *C. zucchellii* IMA088A (violet) are disposed in columns; replicates are indicated with suffix numbers 1, 2 and 3. Compound classes are indicated in different colors and percentages of each class are reported on the right side of the figure.

## 4. Discussion

### 4.1. Molecular Diversity

In the best concatenated 18S rDNA, *rbc*L and *psb*C gene phylogeny (Figure 2), the *Craspedostauros* clade consisted of eight separate taxa including: *Craspedostauros alatus*, *C. alyoubii*, *C. amphoroides*, *C. danayanus*, *C. macewanii*, *C. paradoxus*, *C. ineffabilis* and *C. zucchellii*. The members of the genus *Craspedostauros*, for which the molecular data were available, showed a widespread distribution and habitats. *C. alatus*, *C. danayanus* and *C. macewanii* were growing attached to numerous sea turtles and sea turtle-associated barnacles. *C. alatus* (HK448) was found on the carapaces of several loggerhead sea turtles sampled at the Marine Turtle Rescue Centre in Pula, Croatia, *C. danayanus* (HK637) was collected from the barnacle *Platylepas coriacea* growing on a leatherback sea turtle in Mabibi Beach, Elephant Coast, South Africa (27°21′30″ S, 32°44′20″ E), while three strains of *C. macewanii* (HK634, HK635, HK636) were isolated from the carapace of a captive juvenile loggerhead sea turtle in uShaka Sea World, Durban, South Africa (29°52′02.79″ S, 31°02′45.29″ E). *C. amphoroides* (HK447) was isolated from Herring Cove, Nova Scotia, Canada in the Atlantic Ocean, while *C. alatus* (HK448) and *C. paradoxus* (HK441) were isolated from the Pacific Ocean, respectively, in the Pacific Equatorial upwelling zone (−0.7475° N−−126.03° W) and in Gab Gab Beach, Guam, USA. *C. alyoubii* (UTKSA0083) was collected in Duba, Saudi Arabia, from the Red Sea. *C. ineffabilis* (IMA082A) and *C. zucchellii* (IMA088A) were the only Antarctic species for which DNA sequences were available on online databases. They were collected, respectively, from Inexpressible Island and near the Italian Research Mario Zucchelli Station (Terra Nova Bay).

No molecular data were available for the other *Craspedostauros* species from the southern hemisphere, such as *C. laevissimus* (West & G.S. West) K. Sabbe and as *C. indubitabilis* (Lange-Bertalot & S.I. Genkal) E.J. Cox, and for the other taxonomically accepted species, namely *C. britannicus* E.J. Cox, *C. capensis* E.J. Cox, *C. decipiens* (Hustedt) E.J. Cox, *C. legouvelloanus* Majewska & Bosak. Only transcriptomic data were available for *C. australis* E.J. Cox. For this reason, we did not include all the known *Craspedostuaros* species in the phylogenetic analysis.

Implementing the dataset of Ashworth et al. 2017 [6] with the 18S rDNA, *rbc*L and *psb*C sequences of *C. alatus*, *C. danayanus*, *C. macewanii*, *C. ineffabilis* IMA082A and *C. zucchellii* IMA088A, the molecular data strongly supported the monophyly for the genus *Craspedostauros* (98% SH-aLRT value, 1 posterior probability, 100% bootstrap). This agreed with the DNA-based phylogenies constructed by Ashworth et al. 2017 and Majewska et al. 2021 [6,8]. In our phylogenetic reconstruction, the Antarctic strains (IMA082A and IMA088A) grouped with *C. amphoroides* CCMP797 and formed a well-supported clade (100% SH-aLRT value, 1 posterior probability, 100% bootstrap); however, IMA082A and IMA088A formed two distinct branches, supporting the hypothesis that the Antarctic strains constituted two new species, here described as *Craspedostauros ineffabilis* sp. nov. and *Craspedostauros zucchellii* sp. nov.

Furthermore, we tested different tree topologies to evaluate alternative hypotheses of the evolution of the order Mastogloiales and quadrate rotae taxa. A tree constrained to monophyly *Craspedostauros* and *Achnanthes* species was not significantly different from our best unconstrained tree. However, the tree constraining all Mastogloiales (*Craspedostauros*, *Achnanthes* and *Mastogloia*) to monophyly was statistically significantly worse than the best tree, as was the tree constraining quadrate rotae taxa (*Craspedostaurus*, *Achnanthes*, *Staurotropis* and *Mastogloia*). These results were consistent with those reported by Ashworth et al. 2017 [6] suggesting that a revision of the order Mastogloiales based on a combination of molecular and morphological data is required.

### 4.2. Morphological Diversity

Strains IMA082A and IMA088A displayed all the distinguishing characteristics of the genus *Craspedostauros*, namely cribrate areolae, a central fascia wider than the associated stauros, numerous girdle bands and two H-shaped chloroplasts located "fore-and-aft" of the central area [6,7]. IMA082A resembled *C. indubitabilis* material from Arctic Ocean for the similar valve outline, the dimensions and the stria density [50] and to *C. indubitabilis* material described by Cox (1988) [51] for the similar valve outline, width, stria density, areola size and average number of areolae pores. However, as some details for morphological identification are missing in these studies, it is difficult to establish the true identity of *C. indubitabilis* [46]. IMA088A resembled to *C. indubitabilis* collected from Chile by Rivera et al. 2011 [53] for the similar valve outline, dimensions, stria density, areola size and average number of areolae pores. Sabbe et al. 2003 argued about the conspecificy of *C. indubitabilis* with *C. laevissimus* [46]. This Antarctic endemic diatom was originally described from the McMurdo Dry Valleys region by West & West 1911 [49] and subsequently reported from other localities in Antarctica [44,45,51,52]. However, *C. laevissimus* is the only truly brackish water *Craspedostauros* species (optimally salinity range of 10–15‰) and does not tolerate freshwater or marine conditions [46,54], in accordance with its occurrence in the Larsemann Hills (one single brackish lake) and Rauer Islands [55]. Watanuki 1979 [54] tested the salt tolerance of *C. laevissimus* isolated from two saline lakes in the Sôya coast (east Antarctica) and reported that *C. laevissimus* isolated from Akebi Lake grew very slowly at a salinity of 25‰, while the growth of *C. laevissimus* from Suribati Lake was inhibited at a salinity of 25‰. Thus, in accordance with this ecological aspect, we can exclude that *C. ineffabilis* IMA082A and *C. zucchellii* IMA088A are conspecific with *C. laevissimus.* Despite the common occurrence of *Craspedostauros*-like diatoms in Antarctic and sub-Antarctic regions, morphological details required to the identification at species level are scarce. Furthermore, materials described as *C. laevissimus* and as *C. indubitabilis* collected in different

localities did not display remarkable morphological differences (Table 2). These variations in shapes and features within these two species could be due to phenotypic plasticity [56]. Principal component analysis results highlighted that *Craspedostauros* species have similar cell morphologies and ecological characteristics, indicating that morpho-ecological characters alone do not allow a clear separation among species within this genus. In this sense, limiting the identification of this group of diatoms to solely morphological observations could lead to the underestimation of the diversity of these photosynthetic organisms. For these reasons we mainly relied on the use of molecular data for the taxonomic identification of the Antarctic isolates IMA082A and IMA088A. In this study, *C. laevissimus* and *C. ineffabilis* were distinguished as different species according to their ecology. However, the future sequencing of *C. laevissimus* from Antarctic brackish lakes may highlight its conspecificity with *C. ineffabilis*. This possibility appears as very improbable according to the difference in their ecology, especially their salt tolerance. Alternatively, *C. laevissimus* might represent the transition state from fresh/brackish water to marine environment, or viceversa.

### 4.3. Chemical Diversity

In IM082A and IMA088A, fatty acyls constituted the main biomarker's family, followed by amino acids and peptides, aromatic secondary metabolites, isoprenoids, alkaloids and polyketides. Particularly, fatty acids and isoprenoids are well-known chemotaxonomic markers commonly used to distinguish different algal classes [57,58], while little is known regarding the possible use of other metabolites as biomarkers at any taxonomic level. These results are consistent with the findings of Marcellin-Gros et al. 2020 [12], which reported polar lipids and pigments as the most abundant and diversified metabolites identified over 12 strains of microalgae. This might be a consequence of the increase in the number of studies concerning the analysis of algal lipidomes [12].

Although a large part of the metabolic fingerprints is almost identical between IMA082A and IMA088A, our results underlined the existence of chemical differences at species level (Figures 6–8). These biomarkers might be used in the phylogenetic context, together with the traditional morphological observations and molecular data, for the identification organisms of undetermined taxonomy. In this sense, metabolomic analysis could be introduced in the future as a novel method to characterize photosynthetic organism as a further step in an integrated polyphasic approach.

### 5. Conclusions

The present work upholds the importance of an integrative approaches combining morphological, molecular and chemical data in the recognition of new species. This could overcome the weaknesses of traditional taxonomy based solely on morphological features, which is particularly critical in lineages including numerous cryptic taxa, providing accurate and comparable data. Our results on two Antarctic strains of *Craspedostauros* demonstrate that the integrative approach can be a valuable method to discriminate between species and classify samples within the same genus. Interspecific metabolic variability seems to be strongly determined by the genetic heritage characterizing each species [59]. In this sense, our results clearly showed that IMA082A and IMA088A represented two taxonomic taxa, based on both their metabolic fingerprints and phylogenetic analysis.

The emerging field of metabolomics appears to be an interesting tool to highlight differences between the similar species of microalgae. These data could provide new insights in the understanding of the evolutionary relationships among diatoms, and more especially among the genus *Craspedostauros*.

**Supplementary Materials:** The following supporting information can be downloaded at: https://www.mdpi.com/article/10.3390/jmse10111656/s1, Figure S1: LC-MS chromatograms of IMA082A (a) and IMA088A (b) extracts and replicates in positive ionization mode. Three independent, biological replicates were analyzed per strain. Figure S2: LC-MS chromatograms of IMA082A (a) and IMA088A (b) extracts and replicates in negative ionization mode. Three independent, biological replicates were analyzed per strain. Table S1: Voucher identity and GenBank accession numbers for the taxa used in DNA dataset. GenBank accession numbers listed in order: 18S rDNA, *rbc*L and *psb*C. Table S2: "Cleaned-up" feature lists of features of IMA082A and IMA088A extracts in positive ionization mode. Table S3: "Cleaned-up" feature lists of features of IMA082A and IMA088A extracts in negative ionization mode.

**Author Contributions:** Conceptualization, R.T., L.C. and I.M.; methodology, L.C., E.M. and A.D.L.; formal analysis, R.T. and L.C.; investigation, R.T. and E.M.; data curation, R.T.; writing—original draft preparation, R.T.; writing—review and editing, E.M., A.D.L., L.C., S.S. and I.M.; supervision, L.C. and I.M.; project administration, L.C., I.M. and S.S.; funding acquisition, L.C., I.M. and S.S. All authors have read and agreed to the published version of the manuscript.

**Funding:** This research was funded by the PNRA project PNRA 16 00120—TNB-code: 'Terra Nova Bay barcoding and metabarcoding of Antarctic organisms from marine and limno-terrestrial environments' (PI: S. Schiaparelli). Funding was also provided by Foundation for Science and Technology (FCT), and the Portuguese National Budget funding (UIDB/04326/2020). LC was sustained by FCT Scientific Employment Stimulus (CEECIND/00425/2017).

**Institutional Review Board Statement:** Not applicable.

**Informed Consent Statement:** Not applicable.

**Data Availability Statement:** The nucleotide sequences were deposited into the NCBI GenBank database under the accession numbers: OP354221, OP354493 and OP354495 (18S rDNA, *rbc*L and *psb*C) for IMA082A and OP354222, OP354494 and OP354496 (18S rDNA, *rbc*L and *psb*C) for IMA088A.

**Acknowledgments:** We wish to thank Federico Zorzi (CEASC, Center for Analyses and Certification Services, University of Padova) for his assistance with the SEM-EDS analysis and José Paulo da Silva (CCMAR, Centre of Marine Sciences, University of Algarve) for his assistance with the UPLC-HR-MS/MS metabolomics.

**Conflicts of Interest:** The authors declare no conflict of interest.

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
