# Peer review of "Molecular, Morphological and Chemical Diversity of Two New Species of Antarctic Diatoms, Craspedostauros ineffabilis sp. nov. and Craspedostauros zucchellii sp. nov."

_jmse, doi:10.3390/jmse10111656_

Round 1

Reviewer 1 Report

This manuscript introduces two Craspedostauros strains from the Antarctic coastline and describes them as novel species on the basis of morphological, molecular and metabolomic data. I have two major concerns with this paper:

1) The authors do not make a strong case for the distinction between C. laevissima and C. ineffabilis, relying on the absence of evidence of coastal populations of C. laevissima, as no significant morphological difference exists between the taxa (the SEMs used in the Figures here could be improved--see below) and no molecular data exist for C. laevissima. With the data provided, it appears just as likely that strain IMA082A represents a coastal population of C. laevissima.

2) While the metabolomic data are interesting, they almost feel "tacked on"--nothing is written in the Introduction to explain the need for metabolome data and very little in the Discussion defends its presence in the manuscript. The authors attempt to use the metabolome data to differentiate between the two strains, but do not explain those differences or suggest specific markers that might be used as diagnostic characters (nor do they explain why the metabolomic differences are not artifacts of harvesting the cultures at different growth stages). The references the authors cite describe differences in metabolomes between algal classes, not species. The metabolomic data could be interesting, but it is not integrated very well into the rest of this manuscript. Perhaps the authors could spend some effort comparing these results to those of other raphid pennate and/or Antarctic diatoms if they want to suggest metabolomic data could be an identification tool?

Specific comments:

Introduction

Lines 49-57: The case for molecular data in phylogenetics here us a bit overstated, particularly with regard to stauros-bearing taxa. Analysis of morphological data demonstrated the non-monophyletic nature of the stauros across raphid diatoms long before Ashworth et al. 2017; not just Cox 1999, where Craspedostauros was erected based on morphological data, but also Cox & Williams 2000 (European Journal of Phycology [v. 35], pp. 273-282), which used a cladistic analysis of morphological characters to show stauros-bearing diatoms are not monophyletic. The “incongruence with morphological data” the Ashworth et al. 2017 molecular phylogeny showed was less about the non-monophyletic nature on the stauros and more about the position of Craspedostauros and Staurotropis within the Bacillariales—diatoms with canal raphes.

Line 65: delete “M.P. Ashworth” after Staurotropis

Line 61-61: The Majewska et al. 2020 manuscript had a larger taxon sampling around the Craspedostauros clade than Ashworth et al. 2017—why not use the Majewska et al. dataset as the basis for this analysis?

Line 73: The metabolic fingerprinting aspect of this manuscript feels somewhat disconnected from the rest—there is little in the Introduction to explain the importance of metabolic data to the identity, evolution or classification of Craspedostauros and the stauros-bearing diatoms. What insights do the authors hope to gain and what questions will be answered with the addition of metabolic data to this study system?

Methods and Materials

Lines 121-122: The 18S data were not aligned by their predicted secondary structure? Much has been written about the effects of secondary structure alignment on the resulting diatom molecular phylogeny—this should be addressed.

Lines 137-139: Was the “burn-in” rate arbitrary? How was it determined to discard the first 12500 trees?

Lines 158-159: Cell cycle stage at harvest for metabolomics? Both strains at exponential growth or stationarity phases?

Results

Line 220: What does this mean—that the Antarctic species sequenced in this manuscript are not assigned to species based on DNA data? IMA088A is sister to C. amphoroides—why is this not also C. amphoroides based on DNA data?

Lines 250-251: raphid pennate taxa based on DNA data as well…

Lines 252-274: Why are the morphological descriptions not part of the taxon descriptions?

Line 255: Is there an internal view which is not partially obscured by the valvocopula?

Line 262: 31-58 stria per 10 µm is a large range…is there a pattern to this, such as the density being higher at the apices or center? These values correspond to the length variation in the taxon description—which is correct?

Line 271: “…many pored…” Can the authors be more specific here? Does this refer to the pore density or the number of rows?

Lines 282-316: I would very much like to see a “comments” section in these taxon descriptions. In particular, I am interested in how the authors distinguish C. ineffabilis from C. laevissima. Table 2 leaves out some of the characters illustrated in Sabbe et al. 2003, and I am unclear of what distinguishes a “rectelevatum” from the central helictoglossa illustrated in Sabbe at al. 2003—the raphe endings (internal and external) look extremely similar on the SEMs to my eye…

Line 286, 303: “Rudimentary silica flaps”. Where? Does this refer to the external flaps seen over the central raphe endings in some taxa? Because I see no evidence for any sort of external flap over the central raphe endings in Figs 3 and 4 nor in Sabbe et al. 2003 for C. laevissima.

Figure 4: Can we get an internal view in the SEM where the distal raphe ends are not completely obscured by girdle bands?

Lines 282-316: So, to be clear—the only morphological difference between C. ineffabilis and C. zucchellii are the valve length range and the presence of cribra with more than 7 pores along the raphe slits? C. ineffabilis also has variable areolae sizes (see Figure 3g—striae adjacent to fascia/stauros). Again, a written morphological comparison would be useful in diagnosing these species from each other and from C. laevissima.

Discussion

Line 401: What do the authors mean by “…two distinct leaves”? The Antarctic species are in a clade, but not to the exclusion of C. amphoroides. How confident are the authors in the taxonomic identity of CCMP797? Would a tree constraining the Antarctic strains to monophyly be significantly different than the “best” tree presented here?

Line 420: Be specific about the “crucial details” missing—I do not agree with the characterizations in Table 2 based on the material illustrated in Sabbe et al. 2003, so what characters are the authors using to distinguish C. indubitabilis from their taxa described here? The authors mention many overlapping characters (outline, width, stria density and areolae characters), so what character is “important” enough to distinguish these taxa when the two Antarctic strains presented here are so morphologically similar?

Line 428-429: “optimally salinity range” does not necessarily equal “does not tolerate”. Was C. laevissima tested in culture at different salinity ranges? Were C. ineffabilis and C. zucchelli tested in culture at different salinity ranges? Watanuki 1979 refers to Tropidoneis laevissima as “halophilic”—this does not necessarily equate to “does not tolerate” marine salinities…Many brackish water diatoms can tolerate seawater. What is the most parsimonious conclusion—that C. ineffabilis and C. laevissima are morphologically identical but different species, or that C. laevissima has a range that extends from brackish to coastal environments?

Line 437: Wouldn’t “phenotypic plasticity” suggest that C. laevissima and C. ineffabilis are conspecific? And how would “misclassification” come into play in this scenario—which reference do the authors think misclassified these diatoms?

Line 440-441: How could the authors claim to rely on “molecular data for the taxonomic identification” of these strains when so many of the Craspedostauros taxa to which they might compare these strains have no sequence data? How can the authors suggest that C. laevissima and C. ineffabilis are two distinct species on the basis of molecular data when C. laevissima has no molecular data published?

Line 448-449: “…well-known chemotaxonomic markers.” The references cited here both utilize fatty acid profiles to differentiate between taxonomic classes of algae, not species-specific differentiation.

Line 451: Wouldn’t variation in the metabolomic profile also exist between the strains if they were harvested at different growth stages (exponential growth vs. stationarity)?

Line 452-454: How, exactly, does this study demonstrate how metabolic biomarkers can be used for the identification of organisms of undetermined taxonomy, when this study only compared the metabolomes of two closely-related diatom strains. How would one use the metabolome—what characters would the authors suggest—to determine these diatoms were in the genus Craspedostauros, or even a raphid pennate diatom?

Reviewer 2 Report

Referee report of jmse-1971916 Molecular, Morphological and Chemical Diversity of two new species of Antarctic Diatoms, Craspedostauros ineffabilis sp. nov. and Craspedostauros zucchellii sp. nov., by Riccardo Trentin et al.

In this study the authors describe two new species of Craspedostauros from Antarctic waters. They provide morphological descriptions of the two strains and results of a phylogenetic analysis based on several concatenated gene markers. The authors use a dataset of Ashworth et al. 2017 including three marker gene regions of a huge number of pennate diatom species including Craspedostauros species, to infer a phylogeny with the sequences of their two strains added. The authors conclude that these two strains belong to two distinct species. There is also a report on the metabolome of the two strains. The work is carried out well, objectives are explained clearly, M&Ms are thorough, Results well-presented.

I am not an expert on Metabolomics and therefore refrain from detailed comments on its details.

Introduction: Lines 55-62: The authors state that Ashworth et al. tested if a cladogram inferred from morphological characters and their states showed incongruencies with a molecular phylogeny, and “that we followed this approach” but I see no description of a cladistic analysis of morphological characters in M&Ms.

M&Ms: all clear. I do not comment on metabolic data because I feel not expert enough.

Results: Fig. 2 The circle tree needs to be projected much larger. One does not see the details. With the tree presented as normal phylogram, there is no need for the coloured ranges because taxonomic affiliation can be written to the right.  Do not forget to adjust the scale and mention what it represents.

Photos are of good quality. In the morphology text I missed info about cell size. In Fig. 3a. I notice also two distinct cell lengths over the apical axis, as if the material is not from a monoclonal culture or as if the initial cell size has been restored through sexual reproduction or another form of enlargement.

Try to condense table 2, line distance 1

May I suggest organising Table S1 not by alphabetical order but in order in which the taxa appear in the tree with the colour codes and taxonomic group as indicated in the tree. This way there is additional info corresponding with the tree.

I did refrain from commenting on metabolic data (see above).

Discussion:

Lines 366-385 are basically a summing up of sites/habitats where Craspedostauros specimens have been collected, and that this work adds two specimens, belonging to two species, from Antarctic waters. This mere info can also be found in the big Table. Not clear to me what is Discussion. Given the statement in lines 410-412 that the addition of the two specimens does apparently not change anything in the conclusions already drawn by Ashworth et al. I wonder what is really new here.

The reasoning in lines 414-441 is not entirely clear to me. For instance, how can two markedly genetically and morphologically distinct species […] C. ineffabilis IMA082A and C. zucchellii IMA088A [both be] conspecific with C. laevissimus. ? I understand from the text that C. laevissimus, for which no molecular data exist … apparently, is a brackish water species, which has only been found in brackish lakes in Antarctica. It might be that these lakes are up until now the only places where taxonomists have looked. Now, the authors have looked in the sea, and there they found two strains of Craspedostauros. So, the question that the reader wants to see answered, ‘In how far do the newly described species resemble earlier described ones, and could the specimens the authors describe here be synonyms.  This needs to be addressed head-on. From the text in 414-441 this is not at all clear.

The discussion of the Metabolomes is surprisingly short and seems not overly informative, but this may be a consequence that not very much else is known for the genus. I would expect more reflection on the fact that we have here two representatives of frigid waters whereas all the others seem to occur in balmy environments.

Conclusions is a bit of pushing open an open door. It is already very well known that cryptic diversity is common and – by definition – is not detectable with morphology.

Another concern I have is that the whole study is based on one strain for each of the two species. This is not a problem for the molecular data but it might be for the morphological data. Here there were two very different size classes in the culture of one of the two species, so it might have been worthwhile to explore if the morphological traits are stable in these different size classes. Regarding the metabolome, would it not be better to study this with multiple strains per species?

Minor remarks

The English may need some editing (for example lines 36-39: “The combination [singular] of …, do [plural] not only …  among diatom [no s] species,…” and 56: “several problems [with] the phylogeny [of] raphe-bearing diatoms.” Line 157: Sample [no s] preparation. Line 229: “…phylogeny of concatenated ?the? 18S….” Line 410: This [singular] results [plural]. Just to mention a few examples.

Text composition: 39-40: “The functioning of this taxon.” What taxon? Probably Craspedostauros, but it hasn’t been mentioned yet in introduction. And what is the exact meaning of “different shades of photosynthetic biodiversity of two species of diatoms”’ I greatly appreciate creative, pleasant writing in research articles as long as everything has a crystal-clear meaning.

Reviewer 3 Report

Excellent paper. A complete research exercise.

Author Response

Thank you very much.

Round 2

Reviewer 1 Report

I appreciate the author's response to my previous review. While I still do not agree with all of the authors' taxonomic inferences, they have now explicitly stated their reasoning and created a path forward for testing these taxonomic hypotheses--this is unfortunately often left out of diatom diversity studies. I only have a few reservations about this manuscript which must be addressed before publication:

1) Introducing the metabolomics. I think the authors have actually written the perfect introductory paragraph explaining the inclusion of metabolomic data in this manuscript, but unfortunately it is in the Results rather than the Introduction. I would strongly recommend replacing lines 73-84 with lines 534-541.

2) There are conflicting accounts between the revised manuscript and the "response to reviewer" document regarding the construction and alignment of the DNA sequence dataset (see below). This must be resolved.

Specific comments:

Lines 73-84: I appreciate the attempt to explain the presence of metabolomic data here, but this is less of an explanation of “why” and more a description of what was done. There is the kernel of a good introduction in the statement “…provide information on the functioning of different diatom taxa”. What is meant by “functioning?” This is kind of a vague term, and how would functional differences between diatom taxa be detected better by metabolomics versus DNA sequence data? It might help to explicitly state the goal of evaluating metabolomic characters for taxonomic delimitation, as this paragraph infers. I would also rethink how the Huseby et al. 2012 citation is used in this paragraph. Rather than placing it after the vague “functioning” statement, it would be better served after an explicit statement about the use of metabolomic profiles for taxon delimitation, as Huseby et al. 2012 suggest metabolomic differences reinforce morphological and genetic implications for species diversification between Chaetoceros socialis populations.

Line 162: This doesn’t agree with what was written in the “response to reviewer” document. This statement suggests that the DNA sequence data from each gene was aligned independently using the referenced programs. The “response to reviewer” document suggests that the new sequence data were aligned directly to the Ashworth et al. 2017 dataset, without running SSUalign with the new sequence data. This is problematic, as several Craspedostauros 18S sequences have large introns which were identified and masked from the Ashworth et al. 2017 dataset for phylogenetic analysis. Please explicitly resolve this conflict in the final text.

Line 327: Craspedostauros misspelled.

Line 354-355: “…some morphological resemblance…” That’s quite the understatement…

Line 362: “Silica flaps absent at proximal (or central, depending on your preferred terminology) raphe ends”. Siliceous flaps actually do still obscure the external terminal raphe ends.

Line 465: I would suggest “…two distinct branches…” rather than “lineages”, as both strains share a very recent common ancestor.

Line 465-466: I understand the gist of the argument here, but I would caution the authors about this phrasing in the future. The simple fact that the two IMA strains occupy distinct branches within a clade (in addition to C. amphoroides) does not alone “support the hypothesis that the Antarctic strains constituted two new species”. Three conspecific strains could be represented by three branches in a clade depending on the resolution of the data type used (for example, see Psammodictyon constrictum in Figure 2—two distinct branches for conspecific strains). It is the combination of the distinct branches in the molecular phylogeny, the branch lengths and the morphological data which all contribute to the inference that the three strains represented in your designated clade are different species. The tree topology alone could be inferred to represent several different taxonomic decisions without any additional context.

Line 514-519: Thank you for this. I still do not agree with your taxonomic interpretation presented here, but now there is an explicitly-stated hypothesis that can be tested going forward. That is good taxonomy.

Line 534-541: Good—this is a good explicitly-stated goal which should have been included in the Introduction to explain the presence of metabolomic data. I would replace lines 73-84 with this exact paragraph.

Reviewer 2 Report

The conclusion to describe your strain as a species different from a morphologically apparently identical one collected by others is based on differences in salinity requirements of the strains. So, you use an ecological definition of species. With the added disclaimer in the Discussion 514-519 I am fine with that conclusion. The assumption is: strains growing in different salinity ranges are different species.

I maintain that the presentation of the figures can be improved, especially the letter sizes. I needed a magnifying glass, even at 200% page size. It doesn't take a lot of effort. The circle tree in Fig. 2 can be projected much larger without enlarging the colour bars. Also, the lines in the dendrograms of Figs 8 and 9 are not continuous.
